# Decoding type 2 diabetes mellitus genetic risk variants in Pakistani Pashtun ethnic population using the nascent whole exome sequencing and MassARRAY genotyping: A case-control association study

Asif Jan[1]◉*, Zakiullah[1]◉*, Sajid Ali[2‡], Basir Muhammad[3‡], Amina Arshad[4‡], Yasar Shah[5‡], Haji Bahadur[6‡], Hamayun Khan[1‡], Fazli Khuda[1‡], Rani Akbar[5‡], Kiran Ijaz[6‡]

1 Department of Pharmacy, University of Peshawar, Peshawar, Pakistan, 2 Department of Biotechnology, Abdul Wali Khan University, Mardan, Pakistan, 3 Atomic Energy Cancer Hospital, Swat Institute of Nuclear Medicine, Oncology & Radiotherapy, Swat, Pakistan, 4 Rashid Latif College of Pharmacy, Lahore, Pakistan, 5 Department of Pharmacy, Abdul Wali Khan University, Mardan, Pakistan, 6 Institute of Pharmaceutical Sciences, Khyber Medical University, Peshawar, Pakistan

◉ These authors contributed equally to this work.
‡ SA, BM, AA, YS, HB, HK, FK, RA and KI also contributed equally to this work.
* Zakiullah@uop.edu.pk (ZU); Asif.research1@gmail.com (AJ)

## Abstract

Genome-wide association studies have greatly increased the number of T2DM associated risk variants but most of them have focused on populations of European origin. There is scarcity of such studies in developing countries including Pakistan. High prevalence of T2DM in Pakistani population prompted us to design this study. We have devised a two stage (the discovery stage and validation stage) case-control study in Pashtun ethnic population in which 500 T2DM cases and controls each have been recruited to investigate T2DM genetic risk variants. In discovery stage Whole Exome Sequencing (WES) was used to identify and suggest T2DM pathogenic SNPs, based on SIFT and Polyphen scores; whereas in validation stage the selected variants were confirmed for T2DM association using MassARRAY genotyping and appropriate statistical tests. Results of the study showed the target positive association of rs1801282/*PPARG* (OR = 1.24, 95%CI = 1.20–1.46, P = 0.010), rs745975/*HNF4A* (OR = 1.30, 95%CI = 1.06–1.38, P = 0.004), rs806052/*GLIS3* (OR = 1.32, 95%CI = 1.07–1.66, P = 0.016), rs8192552/*MTNR1B* (OR = 1.53, 95%CI = 0.56–1.95, P = 0.012) and rs1805097/*IRS-2* (OR = 1.27, 95%CI = 1.36–1.92, P = 0.045), with T2DM; whereas rs6415788/GLIS3, rs61788900/*NOTCH2*, rs61788901/*NOTCH2* and rs11810554/*NOTCH2* (P>0.05) showed no significant association. Identification of genetic risk factors/ variants can be used in defining high risk subjects assessment, and disease prevention.

**Data Availability Statement:** All relevant data are within the paper and its Supporting Information files.

**Funding:** This research project was approved and supported financially by Higher Education Research Department, Government of Khyber Pakhtunkhwa Pakistan (Fund No. PMU1-22/ HEREF/2014-15/ VolIV/3408) for laboratory materials and reagent kits. However, the funders had no role in study design, data collection and analysis, decision to publish, or preparation of the manuscript.

**Competing interests:** The authors have declared that no competing interests exist.

## Introduction

Type 2 Diabetes Mellitus (T2DM) characterized by persistent hyperglycaemia is the most frequent subtype of diabetes accounting for around 90–95% of all diabetes cases [1, 2]. It arises from the combination of in-sufficient insulin secretion and excessive secretion of glucagon in a context of insulin resistance. These abnormalities results from alteration in number and/or function of β- and α-cells of pancreas [3, 4]. Loss of pancreatic β-cells in T2DM is believed to occur via apoptosis [5] and autophagy [6]. In addition to the common theme of beta cells failure and peripheral insulin resistance, complex interplay of genetic and non-genetic factors also contribute to the underlying pathophysiology of T2DM [7–9]. Molecular biology investigations have linked genetic variants in number of genes with T2DM. Some of these include *PPARG* [10], *HNF4A* [11], *GLIS3* [12], *MTNR1B* [13], *IRS-2* [14], *NOTCH2* [15], *WFS-1* [16], *GCK* [17], *GCKR* [18], *NEUROD1* [19], *HHEX* [20], *SLC30A8* [21], *VEGFA* [22] and *CAP10* [23]. Other major players involved in the progression of T2DM includes obesity [24], sedentary lifestyle [25], poor diet [26] the metabolic [27] and environmental [28] factors. Identification of these risk factors greatly helps in diabetes assessment and prevention.

The global burden of diabetes has significantly increased and will continue to soar in next few decades [29]. According to International Diabetes Federation IDF Diabetes Atlas 10[th] Edition; in the year 2021 approximately 537 million adults (of age 20–79 years) were estimated to have diabetes. This number is predicated to increase to 643 million by 2030 and 784 million by 2045. A high proportion (81.6%, 432 million) of peoples with diabetes live in low and middle income countries [30]. Prevalence of diabetes is not uniform; South Asian (people residing in China, India, Pakistan Sri Lanka, Bhutan, Nepal, and Maldives) are at high risk compared to other ancestral groups [31, 32]. In the year 2019 number of diabetic patients in Pakistan were 19.4 million in future it is predicated to rise to 34.4 million by 2030 and 37.1 million by 2045 [33]. Pakistan ranked 3[rd] in diabetes prevalence race following China and India [30]. The main factors for marked increase in the incidence of diabetes in Pakistan includes high degree of urbanization and rapid transition in lifestyle [34, 35].

Recent genome wide association studies (GWAS) have greatly increased the number of T2DM associated risk variants but most of these studies have focused disproportionately individuals of European origin. There is scarcity of genomic research in developing countries including Pakistan. Increased prevalence of T2DM among Pakistanis and lack of genetic studies prompted us to design this case-control study. The study aimed to investigate Pashtun ethnic population of Khyber Pakhtunkhwa for T2DM risk variants using nascent technology of Whole Exome Sequencing (WES) and MassARRAY genotyping.

## Materials and methods

### Study subjects

A two stage (i.e. the discovery stage & the validation stage) case-control study have been designed. First Whole Exome Sequencing (WES) was performed to identify and suggest T2DM pathogenic variants in the target population. In the second stage the WES suggested pathogenic variants were confirmed for its association with T2DM using massARRAY genotyping. A total of 1000 individuals (healthy volunteers = 500 and T2DM cases = 500) of Pashtun ethnicity were recruited from seven districts (Peshawar, Mardan, Charsadda, Bannu, Kohat, Dir and swat) of Khyber Pakhtunkhwa, Pakistan for analysis. The cases were matched up with age, gender and ethnicity. Written informed consent was obtained from all the participants. For illiterate/un-educated patient's understanding the informed consent form was read and explained in local Pashtu language and after patient's agreement signed on his/her behalf

by any of his/her relative/attendant. Patient's detailed demographics and clinical parameter were noted on carefully designed proforma. The inclusion criteria for case group were (i) Diabetes diagnosed according to Internal Diabetes Federation (IDF) standard protocols, i.e., Fasting Blood Glucose (FBS) level greater than 126 mg/dL and random blood glucose (RBS) level greater than 200 mg/dL; (ii) Age between 30–80 years and (iii) Belonging from Pakistani Pashtun population. Exclusion criteria for case group were (i) Patients with chronic illness (presence of malignancies), recent severe infection (Hepatitis/Corona virus infection); and (ii) Patients of age not in range of 30–80 years. Whereas Controls were healthy volunteers from general population; age and gender wise matched with case group and fasting blood sugar level less than 100 mg/dL.

## Ethics statement

The study was approved by the ethical committee Department of Pharmacy, University of Peshawar (Approval No. 907/PHAR). All the procedures were carried out in light of Helsinki declaration (1975).

## Blood sampling

Three millilitre whole blood was collected by a trained nurse following aseptic procedures from the median cubital vein of study individuals in EDTA tubes (properly labelled), and was stored at -10˚C.

## DNA extraction and quantification

DNA was extracted from 200 micro litre (μl) whole blood samples of type 2 diabetes patients using Wiz-Prep DNA extraction kit (Wiz-Prep no. W54100) following manufacturer's guidelines. Quantification was done using Invitrogen Qubit™3 and the final DNA concentration was adjusted to 5 ng/μL.

## DNA samples pooling

DNA samples were pooled according to previously described protocol [36, 37]. Two DNA pools one of 500 T2DM cases and second of 500 control subjects were constructed. Each pool containing an equimolar amount of DNA (10ng) from each individual. DNA pooling simplifies sequencing process and reduces cost and time.

## Whole exome sequencing

Whole Exome Sequencing (WES) was carried out at Genomic lab, Rehman Medical Institute (RMI), Hayatabad, Peshawar. Paired end libraries of pool samples were prepared using Illumina Nextera XT DNA library preparation kit. Quantified DNA libraries were sequenced using HiSeq2500 sequencing machine (Illumina, San Diego, CA, USA).

## Bioinformatics analysis of WES data

We employed a custom-built in-house next generation sequencing bioinformatics pipeline to move from raw sequencing data to final variant call. FASTQ files produced by the Illumina HiSeq2500 were filtered to separate low quality reads (Q-score>30) using Trimmomatic software tool [38]. Filtered reads were then aligned to the reference genome (hg19/GRCh37) using a novel logarithmic tool the RAMICS [39]. Variant calling was performed using Genome Analysis Toolkit (GATK) and SAM tools [40, 41]. ANNOVAR was used for variants annotation [42]. The resulting annotated variant list generated by ANNOVAR was stored as Comma-Separated

Values (CSV-file) having separate column for each annotation. The resultant CVS file was loaded/copied into an excel file for easy filtering, viewing and interpretation of data.

## Filtration of WES data

WES generates huge data (>600 GB of data) handling of which is challenging and time consuming. To narrow down the list of variants of our interest the data was filtered as follows. Briefly the annotated file was first manually curated to shortlist putative variants. Non-synonymous, missense variants in exonic and splicing sites were retained, whereas Salient/synonymous variants were discarded (S1 Table). The resultant file was then filtered for selected genes previously reported for T2DM association (S2 Table). Next from list of selected T2DM associated genes we chose *PPARG*, *HNF4A*, *GLIS3*, *MTNR1B*, *IRS-2*, and *NOTCH2* to be further investigated for T2DM susceptibility in the target population.

## Confirmation of WES findings

WES suggested pathogenic variants were cross confirmed for its association with T2DM using Sequenom MassARRAY genotyping (Agena Bioscience, San Diego, CA). MassARRAY of Agena Biosciences easily genotype tens to hundreds of SNPs with great accuracy in short span of time and readily used in mutation detection and genotyping.

## Statistical analysis

Statistical analysis was performed using IBM SPSS (Statistical Package for Social Sciences version 24). Key variables selected for analysis were gender, age, weight, geographical area (districts from where study subjects belongs), smoking, life style, exercise, diet, occupation, and variants reported in selected genes. All variant were tested for Hardy Weinberg equilibrium (HWE) using chi-square ($\chi^2$) test. The difference in distribution of Minor allele frequencies (MAFs) between diabetic patients and control volunteers were determined using $\chi^2$ test. Association between Variants×T2DM were checked using binary logistic regression. A probability value of $p < 0.05$ was considered significant.

# Results

## Subject description

Details of socio-demographics and general characteristics of the study participants are given in Table 1. Prevalence of co-morbidities (i.e. hypertension, renal failure, hypercholesterolemia and retinopathy) was observed higher in diabetic cases compared to controls. Majority of the patients were physically inactive and were from urban areas of Khyber Pakhtunkhwa. Drug and diet compliance were recorded poor in study participants.

## WES results

Exome sequencing identified a total of n = 1248875 SNPs. Among these 691223 were heterozygous, 407572 homozygous, 74390 insertions and 99392 were deletions; 50280 were exonic SNPs, 7910 missense variants and 1987 variants were expressed in pancreas; A total 650 SNPs were reported pathogenic.

## WES suggested T2DM associated SNPs

Exome Sequencing identified a total of n = 9 SNPs (rs1801282/*PPARG*, rs745975/*HNF4A*, rs6415788/*GLIS3*, rs806052/*GLIS3*, rs8192552/*MTNR1B*, rs1805097/*IRS-2*, rs61788900/

**Table 1. Socio-demographic characteristics of cases and controls.**

| Variables | Case n(f) | Control n(f) | P-value |
|---|---|---|---|
| **Gender** | | | 0.897 |
| Male | 358 (71.6%) | 356 (71.2%) | |
| Female | 142 (28.4%) | 144(28.8%) | |
| **Mean age (yrs)** | 57±12.40 | 57±13.43 | 0.951 |
| **Mean weight (kg)** | 61.64±6.07 | 59.55±8.32 | 0.801 |
| **Occupation** | | | 0.589 |
| Labour | 140 (28.0%) | 119 (23.8%) | |
| Govt servant | 30 (6.0%) | 40 (8.00%) | |
| Business man | 10 (2.00%) | 08 (1.60%) | |
| Farmer | 20 (2.00%) | 90 (18.0%) | |
| House wife | 130 (26.0%) | 142 (28.4%) | |
| Driver | 80 (16.0%) | 40 (8.00%) | |
| shopkeeper | 90 (18.0%) | 61(12.2%) | |
| **Geographical area (District)** | | | 0.439 |
| Peshawar | 150 (30.0%) | 139 (27.8%) | |
| Charsadda | 70 (14.00%) | 82 (16.4%) | |
| Swat | 19(3.80%) | 11 (2.20%) | |
| Dir | 11 (2.20%) | 09 (1.80%) | |
| Mardan | 120(24.0%) | 150 (30.0%) | |
| Kohat | 15 (3.00%) | 08 (1.60%) | |
| Bannu | 25 (5.00%) | 19 (3.80%) | |
| Behlola | 90(18.00%) | 82(16.40%) | |
| **Family history of T2DM** | | | 0.02 |
| Yes | 475 (95.0%) | 55 (11.00%) | |
| No | 25 (5.00%) | 445(89.00%) | |
| **Exercise** | | | 0.11 |
| Non-exercising | 411 (82.2%) | 370 (74.0%) | |
| Walking | 75 (15.0%) | 98 (19.06%) | |
| Jogging | 09 (1.80%) | 22(4.40%) | |
| Gym/Sport | 05 (1.00%) | 10 (2.00%) | |
| **Smoking** | | | 0.62 |
| Cigarette | 113 (22.6%) | 98 (19.06%) | |
| Snuff | 220 (44.0%) | 240 (48.0%) | |
| No-smoking | 167 (33.4%) | 162 (32.4%) | |
| **Diet control/compliance** | | | 0.43 |
| Yes | 290 (58.0%) | 311 (62.2%) | |
| No | 210 (42.0%) | 189 (37.8%) | |

N = number; F = frequency; Yrs = years; Kg = kilogram.

*NOTCH2*, rs61788901/*NOTCH2* and rs11810554/*NOTCH2*) in selected genes of our interest. Among these, 5 variants (rs1801282/*PPARG*, rs745975/*HNF4A*, rs806052/*GLIS3*, rs1805097/ *IRS-2* and rs8192552/*MTNR1B*) were suggested pathogenic and rest of 4 SNPs (rs6415788/ *GLIS3*, rs61788900/*NOTCH2*, rs61788901/*NOTCH2* and rs11810554/*NOTCH2*) were marked non-pathogenic by WES (Table 2).

**Table 2. WES suggested diabetogenic and non-diabetogenic SNPs in the discovery stage.**

| Gene | Variant | SNP ID | Chr position | Variation type | Sift (score) | PolyPhen (score) | HGVSc | HGVSp |
|------|---------|--------|--------------|----------------|--------------|------------------|-------|-------|
| PPARG | C>C/G | rs1801282 | 3: 12393125 | Missense_Variant | Deleterious (0.04) | Damaging (0.85) | NM_015869.4:c.34C>G | NP_056953.2:p.Pro12Ala |
| HNF4A | C>C/T | rs745975 | 20:44406053 | Splice variant | Deleterious (0.0) | Damaging (0.97) | NM_000457.4:c.116-5C>T | ------ |
| GLIS3 | G>G/T | rs6415788 | 9:4118111 | missense_variant | Tolerated (0.91) | Benign (0) | NM_001042413.1:c.1367C>A | NP_001035878.1:p.Pro456Gln |
| GLIS3 | A>G/G | rs806052 | 9:4118208 | missense_variant | Deleterious (0.02) | Possibly damaging (0.15) | NM_001042413.1:c.1270T>C | NP_001035878.1:p.Ser424Pro |
| MTNR1B | G>G/A | rs8192552 | 11: 92702962 | Missense_Variant | Deleterious (0.01) | Damaging (0.71) | NM_005959.3:c.71G>A | NP_005950.1:p.Gly24Glu |
| IRS2 | C>C/T | rs1805097 | 13:11043523 | missense_variant | Deleterious (0.0) | Damaging (0.91) | NM_003749.2:c.3170G>A | NP_003740.2:p.Gly1057Asp |
| NOTCH2 | T>T/C | rs61788900 | 1:120029924 | Missense_Variant | Tolerated (0.06) | Benign (0.028) | NM_024408.3:c.137A>G | NP_077719.2:p.Asn46Ser |
| NOTCH2 | C>C/T | rs61788901 | 1:120572572 | Missense_Variant | Tolerated (0.18) | Benign (0.013) | NM_024408.3:c.112G>A | NP_077719.2:p.Glu38Lys |
| NOTCH2 | G>G/C | rs11810554 | 1:120611964 | Missense_Variant | Tolerated (0.09) | Benign (0.12) | NM_024408.3:c.57C>G | NP_077719.2:p.Cys19Trp |

Abbreviations: chr: chromosome; HGVS: human genome association variation; HGVSc: the HGVS coding sequence name; HGVSp: the HGVS protein sequence. Het: Heterozygous; Homo: homozygous.

## Validation/confirmation of WES suggested diabetogenic SNPs

WES generates accurate and reliable results however possibility of false negative and false positive results exists. In order to validate and reduce chances false-negative and false-positive identification rates; all the selected WES identified (n = 9) SNPs were genotyped using MassARRAY and association analysis was performed using appropriate statistical tests. Genotyping and association analysis confirmed strong association (P<0.05) of WES suggested SNPs with T2DM in the study population. WES suggested pathogenic SNPs (n = 5) showed marked difference in the distribution of minor allele frequencies (MAFs) between T2DM patients and controls. Risk alleles burden was observed higher in T2DM patients compare to controls (Table 3).

## Discussion

It is hypothesized that genetic makeup of Pakistani Pashtun population is different from other sub-populations and has unique cultural practices, social values, life style and behaviours, that make this population suitable for genomic studies. No previous comprehensive genomic study as of the present exists in this ethnic group. We confirmed positive association of genetic variations in *PPARG* (rs1801282), *HNF4A* (rs745975), *GLIS3* (rs806052), *IRS-2* (rs1805097) and

**Table 3. Confirmation of WES suggested diabetogenic variants in the validation stage.**

| Gene | SNP | Alleles (major/minor) | MAF (controls) | MAF (cases) | OR (95% CI) | P-value |
|------|-----|----------------------|----------------|-------------|-------------|---------|
| PPARG | rs1801282 | C/G | 0.21 | 0.25 | 1.24 (1.20–1.46) | 0.010 |
| HNF4A | rs745975 | C/T | 0.30 | 0.36 | 1.19 (1.06–1.38) | 0.004 |
| GLIS3 | rs6415788 | G/T | 0.17 | 0.19 | 1.17 (0.87–1.57) | 0.124 |
| GLIS3 | rs806052 | A/G | 0.10 | 0.14 | 1.32 (1.07–1.66) | 0.016 |
| MTNR1B | rs8192552 | G/A | 0.23 | 0.27 | 1.53 (0.56–1.95) | 0.012 |
| IRS2 | rs1805097 | C/T | 0.30 | 0.35 | 1.27 (1.36–1.92) | 0.045 |
| NOTCH2 | rs61788900 | T/C | 0.40 | 0.39 | 1.09 (0.97–1.22) | 0.242 |
| NOTCH2 | rs61788901 | C/T | 0.22 | 0.20 | 1.15 (0.87–1.53) | 0.191 |
| NOTCH2 | rs11810554 | G/C | 0.30 | 0.29 | 1.07 (0.93–1.22) | 0.235 |

*MTNR1B* (rs8192552) with T2DM. Whereas genetic variants *GLIS3* (rs6415788) and *NOTCH2* (rs61788900, rs61788901 and rs11810554) showed no noticeable association.

The Peroxisome Proliferator Activated Receptor Gamma (*PPARG*) is an important transcription factors that has a key role in regulating adipocyte differentiation and controlling glucose and lipid haemostasis [43]. Genome Wide Association Studies (GWAS) in different ancestries reported that single nucleotide polymorphisms in several candidate genes including *PPARG* increases risk of developing T2DM [44, 45]. Among several T2DM associated risk variants one variant extensively studied in different epidemiological study is rs1801282/ *PPARG* [46]. The present study for the first time in Pakistani Pashtun population reported the association of rs1801282/*PPARG* (C>G) with T2DM. WES marked the variant (also known as Pro12Ala) as deleterious and damaging based on SIFT and PolyPhen (0.0 and 0.85) score; Whereas; genotyping by massARRAY confirmed positive association (P = 0.010). Our study findings were compatible with result outcomes of a number of previously genomic studies conducted in different ethnic populations [47–50].

*HNF4A* in beta cells of pancreas regulates expression of genes involved in insulin secretion and glucose metabolism. Polymorphisms in *HNF4A* affects normal glucose metabolism leading to T2DM [51, 52]. The SNP rs745975 in or near *HNF4A* gene in the study population showed strong association (P = 0.004). In concordance to our study findings a large scale association study conducted in Japanese Population also marked rs745975 polymorphism as potential risk factor for T2DM [53]. Likewise variant rs806052/ *GLIS3* showed positive association (P = 0.016) with T2DM in the study population. In contrast to our findings a study conducted in Caucasian population doesn't detect any association of rs806052/ *GLIS3* with T2DM the possible variation in results are due to population heterogeneity [54]. Similarly we confirmed the association of rs1805097/*IRS-2* (P = 0.045) with T2DM in the study population. Previously a case-control study conducted in Bangladeshi population consist of total n = 231 unrelated Bangladeshi (T2DM cases n = 123 and controls n = 108) reported that the variant rs1805097 in *IRS-2* has significantly associated with T2DM in particularly female patients [55]. It is hypothesized that this variant impairs glucose haemostasis leading to T2DM [56, 57]. Moreover we also reported a pathogenic, non-synonymous, heterozygous variant (rs8192552) in *MTNR1B* gene. A similar study case-control study as of ours in Turkish population confirmed the association of variant rs8192552/*MTBR1B* with obesity and related co-morbidities like Insulin Resistance and hypercholesterolemia [58] The reported variant impair normal blood glucose level causing T2DM [59]. We could not confirm association between rs6415788/ *GLIS3*, rs61788900/*NOTCH2*, rs61788901/*NOTCH2*, and rs11810554/*NOTCH2* with T2DM in the target population. Contrary to our study findings genetic alterations in *GLIS3* and *NOTCH2* confer risk for T2DM in Europeans [60, 61]. The conflicting results are due to population genetic differences.

## Conclusion

The present two stage case-control study, using Exome Sequencing and follow-up MassAR-RAY genotyping of selected genes reported association of polymorphism in *PPARG*, *HNF4A*, *GLIS3*, *MTNR1B*, *IRS-2*, and *NOTCH2* with T2DM in Pashtun ethnic population for the first time. The study identified and confirmed potential T2DM causing risk variants in the aforementioned genes. In view of growing number T2DM cases in Pashtun ethnic population it is recommended that similar studies be conducted on other risk variants for T2DM, which will ultimately help in developing risk variant panel genes for screening and identification of genetically susceptible individuals. It will help in devising life style modification and other such strategies that will reduce burden of this fatal and costly disease.

## Supporting information

**S1 Table. Complete list of putative non-synonymous, missense variants.**
(RAR)

**S2 Table. List of selected genes, previously associated with type 2 diabetes mellitus.**
(XLSX)

## Acknowledgments

we are thankful to control subjects (volunteers) and diabetic patients for agreeing to participate in this study. We are grateful to Dr. Johar Ali, head of Centre of Genomics, Rehman Medical Institute (RMI), Hayatabad, Peshawar and Dr. Muhammad Hussain Afridi, consultant endocrinologist of Hayatabad Teaching Hospital (HMC) Peshawar for their kind collaboration and helping us in collecting patient's demographics and blood samples. We also thank Syed Adnan Ali Haider Statistical Assistant at for at the Centre of Genomics, Rehman Medical Institute his kind efforts in bioinformatics analysis of data.

## Author Contributions

**Conceptualization:** Asif Jan, Sajid Ali, Amina Arshad, Hamayun Khan.

**Data curation:** Asif Jan, Yasar Shah, Rani Akbar.

**Formal analysis:** Asif Jan, Zakiullah, Haji Bahadur, Hamayun Khan, Rani Akbar, Kiran Ijaz.

**Funding acquisition:** Zakiullah.

**Investigation:** Asif Jan, Haji Bahadur.

**Methodology:** Asif Jan, Amina Arshad, Haji Bahadur, Kiran Ijaz.

**Project administration:** Amina Arshad.

**Resources:** Zakiullah, Haji Bahadur.

**Software:** Zakiullah, Yasar Shah.

**Supervision:** Sajid Ali.

**Validation:** Basir Muhammad, Yasar Shah, Hamayun Khan, Kiran Ijaz.

**Visualization:** Basir Muhammad, Yasar Shah, Hamayun Khan, Fazli Khuda.

**Writing – original draft:** Asif Jan, Basir Muhammad, Fazli Khuda, Rani Akbar.

**Writing – review & editing:** Asif Jan, Sajid Ali, Fazli Khuda, Rani Akbar.

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
