## [Decision Letter · Decision Letter 0]

5 Jan 2023

PONE-D-22-30913Decoding Type 2 Diabetes Mellitus Genetic Risk Variants in Pakistani Pashtun Ethnic Population Using the Nascent Whole Exome Sequencing and MassARRAY Genotyping: A Seven Districts Based Case-Control Association Study.PLOS ONE

Dear Dr. Jan,

Thank you for submitting your manuscript to PLOS ONE. After careful consideration, we feel that it has merit but does not fully meet PLOS ONE’s publication criteria as it currently stands. Therefore, we invite you to submit a revised version of the manuscript that addresses the points raised during the review process.

Please submit your revised manuscript by Feb 19 2023 11:59PM. If you will need more time than this to complete your revisions, please reply to this message or contact the journal office at plosone@plos.org. Please include the following items when submitting your revised manuscript:A rebuttal letter that responds to each point raised by the academic editor and reviewer(s). You should upload this letter as a separate file labeled 'Response to Reviewers'.A marked-up copy of your manuscript that highlights changes made to the original version. You should upload this as a separate file labeled 'Revised Manuscript with Track Changes'.An unmarked version of your revised paper without tracked changes. You should upload this as a separate file labeled 'Manuscript'.

We look forward to receiving your revised manuscript.

Kind regards,

Giuseppe Novelli

Academic Editor

PLOS ONE

Journal Requirements:

  "This research project was approved and supported financially by Higher Education Research Department, Government of Khyber Pakhtunkhwa Pakistan (Fund No. PMU1-22/HEREF/2014-15/ VolIV/3408)."

   "No competing interest to declare"

Reviewers' comments:

Reviewer's Responses to Questions

**Comments to the Author**

1. Is the manuscript technically sound, and do the data support the conclusions?

Reviewer #1: Partly

Reviewer #2: Partly

Reviewer #3: Yes

2. Has the statistical analysis been performed appropriately and rigorously? 

Reviewer #1: N/A

Reviewer #2: I Don't Know

Reviewer #3: Yes

3. Have the authors made all data underlying the findings in their manuscript fully available?

Reviewer #1: No

Reviewer #2: No

Reviewer #3: Yes

4. Is the manuscript presented in an intelligible fashion and written in standard English?

Reviewer #1: Yes

Reviewer #2: No

Reviewer #3: Yes

5. Review Comments to the Author

Reviewer #1: The work presents a study on selected genetic variants and their association to the risk of T2DM in a sample of the Pakistani population, with the aim of cut the information gap with other populations already studied. Given the well-known complexity of the genetic mechanisms associated with T2DM, the work details are insufficient. It is necessary: to motivate the reasons for the choice of the selected variants leading to the final list, to compare thi list with the variants of other meta-analyses on different ethnic groups, to avoid the use of the term "pathogenetic" referring to risk variants, to distinguish heterozygous and homozygous variants with relative individual/group clinical effects, to present exhaustive tables with numbers of subjects for each variant (do not refer the reader to S1), to analyze the combined genotypes of the variants among individuals/groups to evaluate synergic effects.

Reviewer #2: The aim of this study is to evaluate in the Pakistan population T2DM risk variants comparing two populations, 500 T2DM (Type 2 Diabetes Mellitus) cases and 500 controls. In discovery stage, Whole Exome Sequencing (WES) was used to identify and suggest T2DM pathogenic SNPs; in the validation stage, the pathogenic SNPs identified were confirmed for T2DM association using MassARRAY genotyping and appropriate statistical tests. They found 5 SNPs associated with an increased risk of T2DM, whereas 3 SNPs were not associated with an increased risk.

I suggest to the authors to implement the English language, there are a lot of grammatic mistakes, long sentences, missing commas, words not really English.

- Page 3 beginning: repeated “it is believed”

- Page 3 at the end: are living with diabetes

- Page 4 pakistan…india

And others!

I suggest explaining the age of controls, not only median value and SD (my idea is that they are not affected by diabetes because too young??) and the specific values of normal blood glucose, in the guidelines they are not specified. Is there someone with IGT and IFG? Remember that blood glucose from 110 to 125 mg/dl is associated with IFG, but you say that controls have blood glucose from 70 to 120 mg/dl.

ANNOVAR or ANNOVER? (page 6)

Sailent variants? (page 7)

Wes results: What are 650 pathogenic SNPs identified? In what genes? The paragraph of WES results is confused.

I should suggest explaining why you considered few genes (only 5) instead of the group of genes well know associated with T2DM cited in the introduction and why and how you have reduced the number of Pathogenic SNPs from 650 to 5.

Because your study is not showing new data (except you have confirmed that in pakistani population you can find the same variants than in other population), you can propose new genes because the SNPs you have identified are in genes well known in other populations

Reviewer #3: Dear Editor,

the article by Asif Jan et al. entitled " Decoding Type 2 Diabetes Mellitus Genetic Risk Variants in Pakistani Pashtun Ethnic Population Using the Nascent Whole Exome Sequencing and MassARRAY Genotyping: A Seven Districts Based Case-Control Association Study" is interesting study, on an ethnic group, Pakistani Pashtun, not yet analyzed for the genetic risk score in Diabetes Mellitus. In the last decades, this ethnic group had a significant increase in the prevalence and incidence of type 2 diabetes mellitus.

The experimental design and the statistical evaluation are well conducted.

However, I need clarification:

1. The reason why the analysis was conducted by using Whole Exome Sequencing and then, only the polymorphisms/SNPs reported in table S1 were analyzed. Probably a specific chip could be created for the reported SNPs. I think that some potential data are missing, not evaluable and quantifiable using this approach.

2. It is possible to obtain, beyond the significance for the association between the identified SNPs and Type 2 Diabetes Mellitus, the increased risk of developing Type 2 Diabetes Mellitus in the population under analysis due to the polymorphisms found.

3. It might be nice to know the association percentage of two or more polymorphisms in the population examined.

Kind regards,

6. PLOS authors have the option to publish the peer review history of their article (what does this mean?). If published, this will include your full peer review and any attached files.

Reviewer #1: No

Reviewer #2: No

Reviewer #3: No

---

## [Author Response · Author response to Decision Letter 0]

12 Jan 2023

EXPLANATORY RESPONSE LETTER

We appreciate you and the reviewers for your precious time in reviewing our paper and providing valuable comments. It was your valuable and insightful comments that led to possible improvements in the current version. The authors have carefully considered the comments and tried our best to address every one of them. We hope the manuscript after careful revisions meet your high standards. The authors welcome further constructive comments if any. Below we provide the point-by-point responses;

JOURNAL REQUIREMENTS:

Response: The needful done and revised as per guidelines

 "This research project was approved and supported financially by Higher Education Research Department, Government of Khyber Pakhtunkhwa Pakistan (Fund No. PMU1-22/HEREF/2014-15/ VolIV/3408)."

Response: The needful done as required

 "No competing interest to declare"Please complete your Competing Interests on the online submission form to state any Competing Interests. If you have no competing interests, please state "The authors have declared that no competing interests exist.", as detailed online in our guide for authors at http://journals.plos.org/plosone/s/submit-now This information should be included in your cover letter; we will change the online submission form on your behalf.

Response : The needful done as required

4. Your ethics statement should only appear in the Methods section of your manuscript. If your ethics statement is written in any section besides the Methods, please move it to the Methods section and delete it from any other section. Please ensure that your ethics statement is included in your manuscript, as the ethics statement entered into the online submission form will not be published alongside your manuscript

Response: The needful done as required

Comments to Authors

1. Is the manuscript technically sound, and do the data support the conclusions?

2. Reviewer #1: Partly

3. Reviewer #2: Partly

4. Reviewer #3: Yes

Response: The abstract and conclusion have been revised and made more coherent and representative of the results. Conclusion is made more clear and explicit and rephrased completely.Hope it will be now upto the mark and satisfaction of the reviewer.

 2. Has the statistical analysis been performed appropriately and rigorously?

 Reviewer #1: N/A

 Reviewer #2: I Don't Know

 Reviewer #3: Yes

Response: Yes performed appropriately and rigorously; it's already approved by reviewer.

3. Have the authors made all data underlying the findings in their manuscript fully available?

The PLOS Data policy requires authors to make all data underlying the findings described in their manuscript fully available without restriction, with rare exception (please refer to the Data Availability Statement in the manuscript PDF file). The data should be provided as part of the manuscript or its supporting information, or deposited to a public repository. For example, in addition to summary statistics, the data points behind means, medians and variance measures should be available. If there are restrictions on publicly sharing data—e.g. participant privacy or use of data from a third party—those must be specified

Reviewer #1: No

Reviewer #2: No

Reviewer #3: Yes

Response: Detailed Manuscript with supplementary data was provided earlier. However as per journal requirement and reviewer suggestion this time (in revised manuscript) exhaustive supplementary table S1 and S2 containing A to Z information is provided. Beside this we have more than 600 GB project data some are machine readable files (like FAST-Q, VCF files) some are in excel and other are TSV files. However relevant data (S1 and S2 tables) are provided as supporting information with revised manuscript. And there is no restriction on our data sharing.

4. Is the manuscript presented in an intelligible fashion and written in standard English?

Reviewer #1: Yes

Reviewer #2: No

Reviewer #3: Yes

Response: The manuscript has been revised thoroughly. Abstract, introduction, discussion and conclusion made more comprehensive, coherent and language has been improved significantly. Changes can be seen in track changes option draft provided.Changes and improvements identified by reviewers in draft have been incorporated.

5. Review Comments to the Author

Reviewer #1: The work presents a study on selected genetic variants and their association to the risk of T2DM in a sample of the Pakistani population, with the aim of cut the information gap with other populations already studied. Given the well-known complexity of the genetic mechanisms associated with T2DM, the work details are insufficient. It is necessary: to motivate the reasons for the choice of the selected variants leading to the final list, to compare thi list with the variants of other meta-analyses on different ethnic groups, to avoid the use of the term "pathogenetic" referring to risk variants, to distinguish heterozygous and homozygous variants with relative individual/group clinical effects, to present exhaustive tables with numbers of subjects for each variant (do not refer the reader to S1), to analyze the combined genotypes of the variants among individuals/groups to evaluate synergic effects.

Response: Thank you very much for your previous comments that helped us to improve and polish this manuscript.

Yes; we agree with the reviewer the present study intends to understand the 1). Shared genetic basis of T2DM and 2) possible heterogeneity between Pakistani Pashtun and other populations. 

Pashtun population was selected because this population is 1). Understudied 2). Genetically unique. We attempted to fill genetic information gap as enough studies are carried out in European population but not in the present population. 

As per reviewer suggestions; this time we provided detailed/ exhaustive supplementary table. That includes all relevant data mentioned by the respected reviewer in his comments (i.e. type of variant : heterozygous or homozygous, chromosome number, read depth, amino acid alterations, Minor allele frequency (MAF), and MAFs status in other ethnic population) 

As for variant selection is concern.

We prefer variants that were 

1) exonic

2) Missense 

3) Non-synonymous 

4) Insertions/deletion 

5) Variants in core genes previously reported in European decent but not in our population to highlight the concept of shared genetic and heterogeneity.

Hope it satisfies the reviewers. Any further suggestion (if any) are welcomed and appreciated.

Reviewer #2: The aim of this study is to evaluate in the Pakistan population T2DM risk variants comparing two populations, 500 T2DM (Type 2 Diabetes Mellitus) cases and 500 controls. In discovery stage, Whole Exome Sequencing (WES) was used to identify and suggest T2DM pathogenic SNPs; in the validation stage, the pathogenic SNPs identified were confirmed for T2DM association using MassARRAY genotyping and appropriate statistical tests. They found 5 SNPs associated with an increased risk of T2DM, whereas 3 SNPs were not associated with an increased risk.

I suggest to the authors to implement the English language, there are a lot of grammatic mistakes, long sentences, missing commas, words not really English. Page 3 beginning: repeated “it is believed” Page 3 at the end: are living with diabetes Page 4pakistan…india

Andothers!

I suggest explaining the age of controls, not only median value and SD (my idea is that they are not affected by diabetes because too young??) and the specific values of normal blood glucose, in the guidelines they are not specified. Is there someone with IGT and IFG? Remember that blood glucose from 110 to 125 mg/dl is associated with IFG, but you say that controls have blood glucose from 70 to 120 mg/dl.

ANNOVAR or ANNOVER? (page 6)

Sailent variants? (page 7)

Wes results: What are 650 pathogenic SNPs identified? In what genes? The paragraph of WES results is confused.

I should suggest explaining why you considered few genes (only 5) instead of the group of genes well know associated with T2DM cited in the introduction and why and how you have reduced the number of Pathogenic SNPs from 650 to 5.

Because your study is not showing new data (except you have confirmed that in pakistani population you can find the same variants than in other population), you can propose new genes because the SNPs you have identified are in genes well known in other populations

Response: Thank you for pointing out English and grammatical mistakes; all spelling and grammatical errors pointed out by your good self have been corrected throughout the manuscript (page 1-25). Also repeated words are removed. Moreover while recruiting study participants (controls and cases) it was well assured that they are age, gender and ethnicity matched. 

Normal blood glucose range is modified as per IDF diagnosis criteria for diabetes.( IDF-T2D-CPR-2017-print.pdf)

Thank for thorough review of our manuscript as your good-self pinpoint mistakes like ANNOVAR or ANNOVER? The correct is ANNOVAR. The name is corrected in the revised manuscript.

WES results/statistics reflects: complete list of variants identified. We simplified the list further: filter out the list that out that complete list who much is missense, who much non-synonymous, how much is insertions/deletions how much are benign and how much are disease causing or pathogenic. When we filter our list for pathogenic so the count was 650. Now these 650 variants are in a number of genes. We reporting these genes in sequences some are reported in this manuscript and remaining will be reported in other articles which are in pipeline and hopefully that will be submitted to the same journal.

In this study we have selected some prominent gene variants for further assay and analysis as identified by WES results.

Ideally this research study first of its kind intended to identify and confirms daibetogenic in Pashtun population. By comparing this data with other ethnic population research data it seems a simple validation study. However even such studies as of present; needed to assess the genetic risk among Pakistanis are lacking. To fill the genetic information gap btw European and Pakistan we design this study. The present study reported variants which are common in our cohort and European cohort representing share genetic basis of T2DM and reported some other variants which are responsible for T2DM in European cohort but not in present population. Which represent the term heterogeneity.

Hope it satisfy your goodself.

Reviewer #3: 

DearEditor,

the article by Asif Jan et al. entitled " Decoding Type 2 Diabetes Mellitus Genetic Risk Variants in Pakistani Pashtun Ethnic Population Using the Nascent Whole Exome Sequencing and MassARRAY Genotyping: A Seven Districts Based Case-Control Association Study" is interesting study, on an ethnic group, Pakistani Pashtun, not yet analyzed for the genetic risk score in Diabetes Mellitus. In the last decades, this ethnic group had a significant increase in the prevalence and incidence of type 2 diabetes mellitus.

The experimental design and the statistical evaluation are well conducted.

However, I need clarification:

1. The reason why the analysis was conducted by using Whole Exome Sequencing and then, only the polymorphisms/SNPs reported in table S1 were analyzed. Probably a specific chip could be created for the reported SNPs. I think that some potential data are missing, not evaluable and quantifiable using this approach.

2. It is possible to obtain, beyond the significance for the association between the identified SNPs and Type 2 Diabetes Mellitus, the increased risk of developing Type 2 Diabetes Mellitus in the population under analysis due to the polymorphisms found.

3. It might be nice to know the association percentage of two or more polymorphisms in the population examined.

Response: Thanks for valuation suggestions and critical analysis.

1: Actually WES has been performed on pools of DNA as described in methodology section and an overall picture of the variants that are present is obtained. We have around 30 GB data generated through this and are analysing and publishing from various aspects.

In current study we have validated some of the associated gene variants, previousely reported and also suggested and supported by our WES results. Validation have shown association of some variants and non association of some variants associated in other ethnicities.

2: Yes we agree and carried out a whole picture irrespective of the previousely reported variants and also some other aspects have been published by our group somewhere else.

Current study is limited to some selected gene variants that we considered more important to be reported here.

3: Your suggestion is valuable. However we think that the current results of the study are sufficient to infere and get the conclusion drawn in this manuscript.

Hope it satisfies the reviewers. Further suggestion (if any) is appreciated.

---

## [Editor Report · Decision Letter 1]

16 Jan 2023

Decoding Type 2 Diabetes Mellitus Genetic Risk Variants in Pakistani Pashtun Ethnic Population Using the Nascent Whole Exome Sequencing and MassARRAY Genotyping: A Case-Control Association Study.

PONE-D-22-30913R1

Dear Dr. Jan,

We’re pleased to inform you that your manuscript has been judged scientifically suitable for publication and will be formally accepted for publication once it meets all outstanding technical requirements.

Kind regards,

Giuseppe Novelli

Academic Editor

PLOS ONE
---

## [Editor Report · Acceptance letter]

19 Jan 2023

PONE-D-22-30913R1 

Decoding Type 2 Diabetes Mellitus genetic risk variants in Pakistani Pashtun ethnic population using the nascent Whole Exome Sequencing and MassARRAY genotyping: A case-control association study. 

Dear Dr. Jan:

I'm pleased to inform you that your manuscript has been deemed suitable for publication in PLOS ONE. Congratulations! Your manuscript is now with our production department. 

Kind regards, 

on behalf of

Prof. Giuseppe Novelli 

Academic Editor

PLOS ONE